# Bacterial Meningitis in Buffaloes in Brazil

**DOI:** 10.3390/ani14030505

**Published:** 2024-02-03

**Authors:** José Diomedes Barbosa, Henrique dos Anjos Bomjardim, Camila Cordeiro Barbosa, Carlos Magno Chaves Oliveira, Paulo Sérgio Chagas da Costa, Carlos Eduardo da Silva Ferreira Filho, Natália da Silva e Silva Silveira, Marcos Dutra Duarte, Luís Antônio Scalabrin Tondo, Marilene de Farias Brito

**Affiliations:** 1Instituto de Medicina Veterinária, Universidade Federal do Pará, Castanhal 68740-970, Brazil; camilabarbosamedvet@gmail.com (C.C.B.); cmagno@ufpa.br (C.M.C.O.); carloseduardofilho.mv@gmail.com (C.E.d.S.F.F.); nataliasilvasilveira1@gmail.com (N.d.S.e.S.S.); duarte.marcos@gmail.com (M.D.D.); 2Faculdade de Medicina Veterinária, Instituto de Estudos do Trópico Úmido, Universidade Federal do Sul e Sudeste do Pará (Unifesspa), Xinguara 68557-335, Brazil; henriquebomjardim@unifesspa.edu.br; 3Faculdade Antonio Leite FAL/UniBTA, Campus Castanhal, Castanhal 68742-000, Brazil; paulocosta4621@gmail.com; 4Departamento de Epidemiologia e Saúde Pública (DESP), Instituto de Veterinária (IV), Universidade Federal Rural do Rio de Janeiro (UFRRJ), Seropédica 23890-000, Brazil; luisantonio.tondo@gmail.com (L.A.S.T.); mfariasbrito@uol.com.br (M.d.F.B.)

**Keywords:** infectious disease, nervous system, *Bubalus bubalis*

## Abstract

**Simple Summary:**

Buffalo farming has become economically important in Brazil given the highly rustic nature of the species and the better nutritional characteristics of buffalo meat and milk. However, the sector has been impacted by the occurrence of different diseases in buffaloes that have been poorly described in the international literature to date. This study reports and describes the clinically observed neurological signs and the macroscopic and microscopic lesions of bacterial meningitis in adult buffaloes raised in the Amazon biome. These lesions are associated with fractures of the base of the horn and exposure of the frontal sinus.

**Abstract:**

Meningitis is the inflammation of the membranes surrounding the central nervous system and is poorly described in water buffaloes. Five cases of meningitis in adults buffaloes of the Murrah and Mediterranean breads were studied. All buffaloes came from a farm located in the municipality of Castanhal, Pará, Brazil at different times. Clinical examination showed neurological clinical signs, such as apathy, reluctance to move, spastic paresis especially of the pelvic limbs, hypermetria, difficulty getting up, pressing of the head into obstacles and convulsion. In three buffaloes, a large part of the horn had been lost, exposing the corresponding frontal sinus, through which a bloody to purulent exudate flowed. The hemogram revealed neutrophilic leukocytosis. At necropsy, adherence of the dura mater to the periosteum and a purulent to fibrinopurulent exudate were observed in the sulci of the cerebral cortex and on the pia mater over almost the entire surface of the brain and throughout the spinal cord. The cerebrospinal fluid had a cloudy aspect with fibrin filaments. The histopathology of buffaloes confirmed the diagnosis of bacterial fibrinopurulent meningitis. Buffaloes are susceptible to bacterial inflammation of the meninges due to fractures of the base of the horn and mostly present with neurological manifestations.

## 1. Introduction

Rabies and botulism are the two main causes of death in cattle in Brazil, and these conditions result in neurological and motor signs [1,2,3]. In Brazil, many diseases of the nervous system have already been described in cattle [3,4]; on the other hand, in buffaloes, there are few studies that address diseases related to this system. In the state of Pará, northern of Brazil, different causes of compressive lesions that affected the central nervous system (CNS) of buffaloes [5] and an outbreak of nerve listeriosis in buffaloes have already been described [6]. Bacterial meningitis is important as the cause of neurological and motor clinical signs in cattle and can lead to significant economic losses for the producer and compromise animal welfare [7].

Meningitis is the inflammation of the membranes surrounding the central nervous system and occurs occasionally in production animals; meningitis can be caused by viral and bacterial infections or fungi on a smaller scale. Among these causes, bacterial meningitis is the most common, and generally more than one type of bacteria is involved and associated with other infectious foci, such as peritonitis, pleuritis, pericarditis, endophthalmitis, polyarthritis and pituitary abscesses [8,9,10,11].

There are four pathways for the entry of microorganisms into the CNS, with the most frequent being the hematogenous pathway; the other pathways include neurogenic invasion through peripheral axons, invasion of the olfactory mucosa and direct dissemination [9,12,13]. The clinical signs that animals with meningitis present with correspond to the region of the affected CNS. When the infection is located in the spinal cord meninges, muscle spasms with stiffness of the limbs and neck and cutaneous hyperesthesia may occur. Signs of irritation, such as muscle tremors and seizure, are associated with cerebral meningitis [9].

Given the different etiologies and clinical signs presented by ruminants affected by diseases of the nervous system, veterinarians should be able to identify them and establish a differential diagnosis [3]. Due to the scarcity of literature on meningoencephalitis in the bubaline species, the objective of this study is to describe the epidemiological and clinical-pathological aspects of 5 cases of bacterial meningoencephalitis in this species.

## 2. Materials and Methods

The study comprised observations performed in five buffaloes (Buffaloes 1 to 5). Epidemiological data, such as age, sex, breed, farm location and type of management adopted on the properties, were obtained at the time of the clinical visit. Buffaloes with clinical neurological signs were submitted to general and specific clinical examinations of the nervous system according to Dirksen et al. [14]. In Buffaloes 2 and 4, a complete blood count was performed according to Schalm and Jain [15].

Necropsy was performed on all 5 buffaloes after natural death. Fragments of all organs of the Buffaloes 2 and 4 were collected and fixed in 10% buffered formalin. These fragments were sent to the “Setor de Anatomia Patológica’’ (Pathological Anatomy Sector) of the “Universidade Federal Rural do Rio de Janeiro” for histopathological examination. The tissue samples were routinely processed, embedded in paraffin, cut at 5 µm and stained with hematoxylin and eosin (H&E). In addition, data from analysis of cerebrospinal fluid (CSF) collected from the cisterna magna of an animal were included for physical evaluation according to Dirksen et al. [14].

## 3. Results

Of the 5 buffaloes, three were females, and two were males. All buffaloes came from a farm in the municipality of Castanhal, Pará state, Brazil. The buffaloes were kept in a semi-extensive breeding system with a herd of 40 buffaloes. Buffaloes 1 and 2 were of the Murrah breed, and Buffaloes 3, 4 and 5 were of the Mediterranean breed, with ages ranging from eight to 20 years. The cases occurred at different times.

On clinical examination, all animals presented a body score between 1.5 and 3 (scale of 1 to 5) and an elevated body temperature, and all showed apathy, inappetence, dehydration, pale mucous membranes, sternal decubitus with the head turned to the flank (Figure 1a), reluctance to move and ataxia of the pelvic limbs. Additionally, Buffalo 2 presented hypermetria and muscle spasms with stiffness in the pelvic limbs. Buffalo 3 had difficulty getting up and seizures and pressed his head against the corral fence. Buffaloes 1, 2 and 5 had horn fractures, with exposure of the underlying frontal sinus and exudation of bloody to purulent material (Figure 1b). The horn fractures in all three cases were caused by fight between the buffaloes.

In Buffaloes 2 and 4, the blood count revealed neutrophilic leukocytosis. In the cerebrospinal fluid analysis of Buffalo 2, a turbid appearance and the presence of fibrin filaments were observed (Figure 2).

At necropsy of Buffaloes 1, 2 and 5, purulent content was observed in the frontal sinus (Figure 3a). In addition, in Buffalo 5, there was loss of bone tissue in the frontal sinus (Figure 3b). In all buffaloes, the dura mater and the encephalic leptomeninges were opaque and thickened. Additionally, in Buffalo 2, the brainstem was covered with fibrinopurulent exudate (Figure 4a), and thickening of the spinal cord meninges with the presence of fibrin filaments was observed (Figure 4b). In Buffaloes 3 and 5, there was a fibrinous purulent exudate between the brain gyri. Peritonitis was observed in Buffalo 4. Increased amounts of peritoneal fluid was observed that was yellowish with filaments, and clots of fibrin adhered to the serous layer of the abdominal organs.

The histopathology of Buffaloes 2 and 4 showed a marked inflammatory infiltrate composed of viable and degenerated neutrophils, as well as a large amount of fibrin in the leptomeninges of the regions of frontal and parietal cortex, olfactory bulb, brainstem and pons (Figure 5a,b). Thrombi consisting of fibrin, neutrophils and lymphocytes, as well as fibrinoid necrosis of the blood vessel wall were present in the leptomeninges. In Buffaloes 2 and 4, basophilic coccobacilli were observed adhering to the leptomeninges and occasionally inside the blood vessels. In Buffalo 4, there were multifocal areas of hemorrhage associated with areas of fibrinoid necrosis of the blood vessel wall.

## 4. Discussion

The diagnosis of meningitis was based on clinical examination, necropsy and histopathological findings. The affected animals were adults over eight years of age, unlike cattle in which the disease is more often reported in young animals [7,9,16].

The origin of meningitis in Buffaloes 1, 2 and 5 was associated with fractures at the base of the horn with exposure of the frontal sinus and consequent bacterial contamination. In Buffalo 4, it was associated with peritonitis. In Buffalo 3, it was not possible to establish its origin. In cattle, meningitis is often associated with umbilical infections in young animals, failure in the transmission of passive immunity through the colostrum, malnutrition and concomitant viral infections [9,17]. These causes differ from the findings from the buffalo in this study, in which horn fractures are also attributed as a cause. Loretti et al. [8], Barros et al. [9], Lemos and Brum [10], and Margineda et al. [11] also report other infectious processes and origins of meningitis, such as peritonitis, pleuritis, pneumonia, enteritis, pericarditis, endophthalmitis, polyarthritis and pituitary abscesses.

All buffaloes exhibited typical clinical signs of CNS involvement. In Buffaloes 1, 3, 4 and 5, the clinical signs presented were exclusively related to the encephalon; however, in Buffalo 2, clinical signs related to the spinal cord were also present. The buffaloes in our study were found to be sternal decubitus with their heads turned toward the flank. The buffaloes exhibited the following features: a reluctance to move, hypermetria, ataxia and muscle spasms with rigidity of the pelvic limbs, difficulty getting up, seizures and pressing of the head against obstacles. In other species, cervical rigidity, hyperesthesia, seizures, pedaling movements, opisthotonos, circling, and vocalization have been reported [8,9,10,18,19].

Barros et al. [9] described the occurrence of neutrophilic leukocytosis in cattle with acute cases of meningitis and leukocytosis with a predominance of monocytes in chronic cases. In Buffaloes 2 and 4 with chronic meningitis, the leukocyte response differed from the response observed in cattle with chronic meningitis. Specifically, the buffaloes exhibited neutrophilia, whereas the cattle exhibited monocytosis. Moreover, in addition to neutrophilic leukocytosis being related to meningitis, it could also be associated with concomitant conditions in these buffaloes, such as the fracture of the horn in Buffalo 2 and peritonitis in Buffalo 4.

The necropsy findings in the present report were similar to those described in cattle with meningitis [9,10,12], goats [18,19] and calves with pituitary abscess syndrome that developed meningitis and exhibited yellowish or whitish, opaque and slightly thickened leptomeninges over the brainstem, cerebellum and cervical spinal cord [8]. Animals with meningitis present turbid, amber cerebrospinal fluid that may contain fibrin filaments [12,19], a finding similar to that observed in Buffalo 2 in the present report. The accumulation of polymorphonuclear inflammatory cells, including some mononuclear cells, and fibrin in the subarachnoid space as observed in the histopathological examination of two buffaloes is similar to that observed in the bovine species [7,10,11,12].

Frontal sinus contamination as a result of a cornual process fracture was found to be the most important focus of bacterial contamination for the development of fibrinous purulent meningoencephalitis in the animals in this study. Bacterial sinusitis can evolve into meningitis through direct extension of the lesion due to the close relationship between the frontal sinuses and the braincase.

Horn fractures predispose buffaloes to meningitis and may be associated with dominance behaviors (corresponding by hierarchy) among animals. In addition, management errors, poorly planned facilities and accidents due to inadequate transport are also related to trauma to the horns [20]. The anatomical shape of the horns of mestizo or pure Murrah buffalo horns are susceptible to accidents between animals at the facilities. Because their horns have a curved shape, mestizo or pure Murrah buffaloes can attach their horns to other animals when fighting, or horns can get stuck in structures in the corrals and fences. These situations favor fractures when buffaloes instinctually try to free themselves.

Horn base fractures commonly causes exposure of the underlying frontal sinuses. Such lesions, if not managed, are a gateway to contamination by environmental bacteria. In these cases, the frontal, cuckold and paranasal sinuses have hemorrhagic walls and may be partially or fully filled with yellowish, greenish or dark (necrotic) and fetid exudate; animals may have bleeding and nasal exudation. Myiasis complicates the picture of sinusitis, and neurological impairment results in meningitis.

In addition to the decrease in bone density due to phosphorus deficiency, pathological fractures can also occur due to proliferative lesions and bone neoplasms. Treatment and control measures, such as surgical dehorning, mopping calves in the first weeks of life and basic care to mitigate management failures, can prevent these accidents from culminating in meningitis.

## 5. Conclusions

The cases of bacterial meningitis in buffaloes are secondary to inflammatory process situated in the frontal sinus resulting mainly from fractures of the base horn or peritonitis, providing evidence for the importance of infectious processes in other parts of the animal’s body in the genesis of meningitis in the species.

## Figures and Tables

**Figure 1 animals-14-00505-f001:**
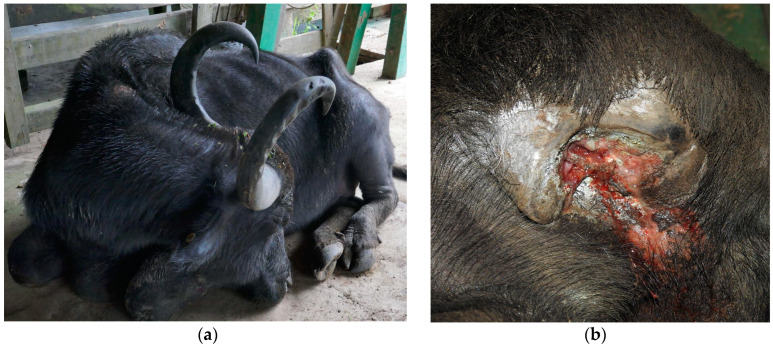
Bacterial meningitis in buffaloes in Brazil; (**a**) Buffalo 4 is in sternal decubitus with the head facing to the left flank and closed eyes; (**b**) In Buffalo 5, the right cornual process is fractured, and a purulent exudate is observed.

**Figure 2 animals-14-00505-f002:**
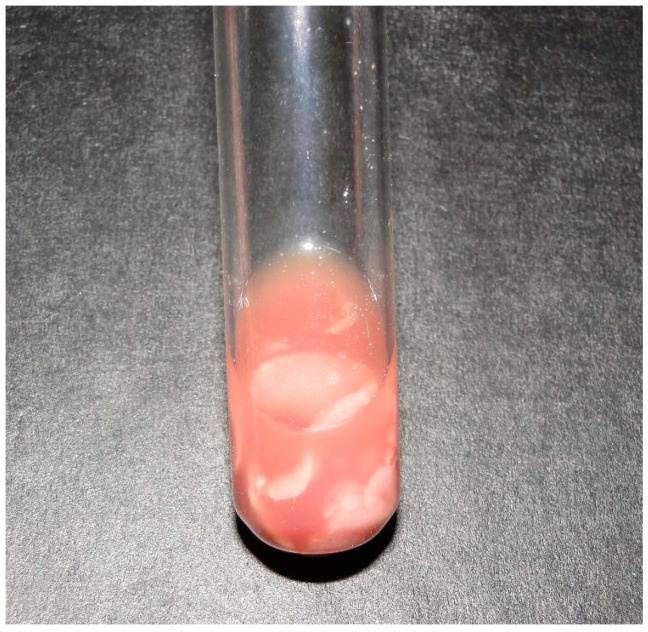
Bacterial meningitis in buffaloes in Brazil. The cerebrospinal fluid from Buffalo 2 appears cloudy and reddish with fibrin clots.

**Figure 3 animals-14-00505-f003:**
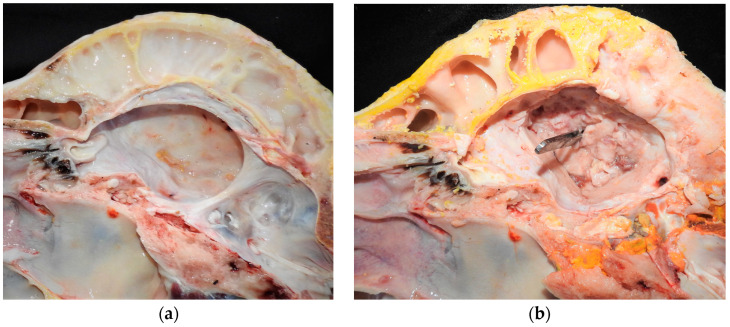
Bacterial meningitis in buffaloes in Brazil; (**a**,**b**) In Buffalo 5, purulent content in the ventral base of the braincase is observed, and passage of anatomical forceps from of the cornual sinus to the braincase demonstrates the potential gateway of the infectious process.

**Figure 4 animals-14-00505-f004:**
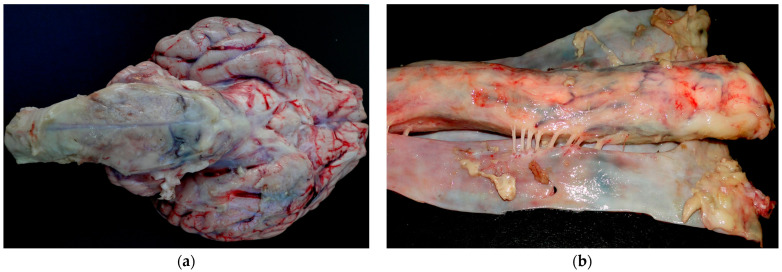
Bacterial meningitis in buffaloes in Brazil; (**a**,**b**) Purulent content in the ventral base of the brainstem and fibrin clots in the subarachnoid space from spinal cord are observed in Buffalo 2. The leptomeninges are opaque and thickened.

**Figure 5 animals-14-00505-f005:**
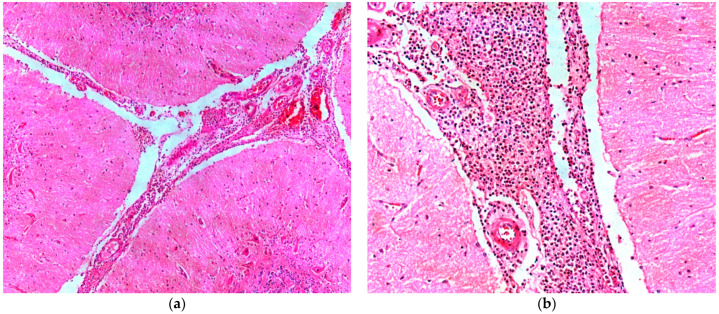
Bacterial meningitis in buffaloes in Brazil. (**a**) In Buffalo 4, a dense neutrophilic and lymphoplasmacytic inflammatory infiltrate in the area located below the meninges with hemorrhagic foci are observed. H&E, obj. 5; (**b**) Buffalo 4, the meninges are thickened by a neutrophilic and lymphoplasmacytic inflammatory infiltrate, and the blood vessels exhibit fibrinoid necrosis. H&E, obj. 20.

## Data Availability

Data are contained within the article.

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
