# Peer review of "Bacterial Meningitis in Buffaloes in Brazil"

_animals, 2024, doi:10.3390/ani14030505_

Round 1
Reviewer 1 Report
Comments and Suggestions for Authors
The manuscript on meningitis in buffalo is both original and holds significant importance, particularly in the context of differential diagnoses for other diseases. I recommend titling it "Post-Traumatic Bacterial Meningitis in Buffaloes" to better convey the focus. Review deeper into the intricacies of frontal sinuses and including information on the species' anatomy that may contribute to brain involvement would enhance the introduction. Additionally, exploring sinusitis, a potentially significant factor, is advisable.
To provide a comprehensive understanding, consider incorporating a schematic representation illustrating how the causative agent reaches the brain. The macroscopic image is good of the brain revealing basilar meningitis involving the brain stem prompts a discussion on the mechanism of its arrival—perhaps influenced by gravity—and whether this could contribute to confusing clinical signs.
Including a photograph of sinusitis is crucial, given its likely importance in the context of the findings . Regarding microscopic photographs, improving their resolution is essential for a clearer representation. It's worth addressing the absence of bacterial isolation in the cases—knowing the specific agent involved, such as Trueperella pyogenes, could significantly contribute to the findings. Consider discussing the implications of not isolating the bacteria and the potential benefits of obtaining this information for a more comprehensive understanding of the disease.
Author Response
Dear Reviewer,
We would like to express our sincere gratitude for the efforts dedicated to reviewing our scientific article entitled "Bacterial Meningitis in Buffaloes in Brazil". Your considerations and suggestions played a fundamental role in improving the grammar and cohesion of the manuscript. We deeply appreciate the thorough and critical analysis you carried out, helping to improve the clarity and overall quality of the work. Your commitment to academic rigor was evident in every comment you provided. We understand the importance and requirement of peer review, and we are aware that the task can be challenging. Therefore, we sincerely thank you for your time and expertise, which were instrumental in raising the standard of our article. This constructive feedback strengthened our work and helped make it more suitable for publication. We hope that our research will be a valuable contribution to the academic community. Once again, we thank you for your dedication to reviewing our article.
Sincerely, the authors.

Reviewer 2 Report
Comments and Suggestions for Authors
For livestock animals to be productive, they must be healthy.
Would it be possible to say how many buffaloes there were on the farm?
Was there a history of cornal problems in the farm's buffaloes? Five sick animals, for the same problem and at the same time, seems like something important.
Was any bacteriological examination carried out to isolate possible etiological agents?
Line 54 - I would suggest mentioning not only economic losses, but also the compromising of the well-being of sick buffaloes. Sick animals suffer a lot.
Line 80 - It is never complete. there will always be something missing. You can always take one more test. That's a bit of logic. However, there is not a big problem.
Line 82- Were the buffaloes euthanized? How? Or did they die naturally?
Line- 115-120 and 149 - In Buffalo meningitis was associated with peritonitis. Did you find peritonitis in buffalo 4?
Line 160 - Were there any signs of peritonitis?
Line 169 - But could there be peritonitis?
Author Response

(The authors gave the same response as above.)

Reviewer 3 Report
Comments and Suggestions for Authors
1. "Abstract" is too long, and it is recommended that the author condense this section. 2.
2. In lines 72-73, the author mentions "epidemiology," but this study is centered on "clinical pathology" and not "epidemiology". The study was not centered on "epidemiology".
3. "Materials and Methods" seems to be too briefly written by the authors, and it is suggested that the authors add the necessary steps. 4.
4. There is no mention of "elevated body temperature" in the "Clinical Symptoms" of the animals, can the authors provide it?
5. The quality of the picture in Figure 6 seems to be less than satisfactory, can the authors replace this picture?
6. Is it possible to standardize the size of the images in Figures 6 and 7?
7. In the "Discussion" section, there are too many paragraphs, and the authors are advised to reorganize the structure.
Author Response

(The authors gave the same response as above.)

Reviewer 4 Report
Comments and Suggestions for Authors
Title of the manuscript: Meningitis in Buffaloes
Aim: to describe the epidemiological and clinical-pathological aspects of meningoencephalitis in this species.
Highlights: the paper describes 5 cases of meningitis in a poor studied animal species and related it to previous conditions.
General comments: In my opinion, the manuscript is interesting but it must be improved. Title is general but the paper just describes one type of exudate related with Gram positive coco-bacilli. Thus, title could be changed to “Bacterial meningitis in buffaloes in Brazil”. The aim is not completely addressed: epidemiologic and pathologic information is scanty. Were cases part of an outbreak or were independent cases? Were all from one or several farms? Age was determinant for the disease?
Authors described 5 cases of meningitis but did not evaluate the potential aetiology. They found gram positive organism but did not perform microbiologic/molecular studies for its identification. It could be due to several reasons, which were not exposed in the manuscript. In spite of that, pathology can be enough for diagnostic but pathologic study might be deeper, i.e: were CNS parenchyma or other organs affected? Aetiology might be suggested based in results and previous reports (Are any frequent microorganism affecting buffaloes for example?)
Methods are not consistent in all cases since some tests were done in one animal. Those test were nor determinant for the diagnosis. The inclusion of those should be better justified in discussion. There are results mentioned in “Discussion” that were not pointed in Materials and Methods or “Results” (i.e: Gram staining).|
Microscopic findings are not well described, and it is important because of the lack of bacteriology or molecular studies. They say one case had peritonitis but did not describe the exudate or opinion about why it could be related to meningitis.
Photos are good. A table for clinic and epidemiologic data could be useful for summarize the information.
Finally, English is not my natural language but I found some terms that should be corrected (i.e. “brain bridge” could be changed by “pons” in line 131). Abstract must be re-evaluated to make it easier to understand.
Specific comments:
Line 82: “Necroscopic examination was performed…” should be changed to “Necropsy was performed…”
Line 129: “composed of neutrophils and degenerates,…” could be “composed of viable and degenerated neutrophils,…”. In case the meaning is other, please rewrite the sentence.
Line 131: “brain bridge” is not an anatomic region. May be authors referred pons, located in ventral side of the rombencephalon.
Line 134: How authors did identify Gram positive bacteriae? It was not mentioned in Methods.
Line 147: Authors indicated animal 3 had horn fracture but in results it was not mentioned. Please, correct that.
Line 149: Peritonitis of animal 4 was not mentioned in necropsy findings.
Line 167-170: Clinic course of disease was not described previously but authors compared chronic and acute meningitis. Please, include epidemiologic data in “Results” for doing that.
Author Response

(The authors gave the same response as above.)

Round 2
Reviewer 4 Report
Comments and Suggestions for Authors
The manuscript has been improved but, in my opinion, there still are some issues that must be corrected or clarified.
Tests performed were not consistent: only necropsy was done in all cases. Fortunately grossly changes were clear and allowed authors to diagnostic, but results of other tests are difficult to understand. I strongly recommend to summarize results in a table.
The course of the disease was not referred until paragraph started on line 175. There, authors evaluated blood results and clinic course. Which was the criterion for course determination? Was it clinic or pathology? Which is the importance of the course? Can that impact on treatment or diagnostic? Authors should evaluate the aim of its analysis since it is in the manuscript. On the other hand, they assigned blood results exclusively to meningitis and forget concomitant disease as horn fracture or peritonitis, which probably impacted on blood leukocytosis. Those conditions should impact on blood results, and that must be analysed.
Paragraph starting on line 80 indicated that animals 2 and 4 were tested by histopathology but line 142 tells all cases showed inflammation of meninges at histopathology. One of two is wrong, or text should be re-written.
Line 59: “There are four pathways for the entry of microorganisms into the CNS, being the most frequent the haematogenous pathway”.
Line 71: “…pathological aspects of 5 cases of bacterial meningoencephalitis in this species”
Line 101: “The horn fractures in all three cases were caused by fighting between the buffaloes.”
Line 147: Authors says there were Gram positive coccobacilli. This is wrong since Gram staining was not performed. So, that can be suspected but not affirmed. Perhaps they can say “basophilic coccobacilli”. Are previous reports which may suggest any potential aetiology? In case, it can be interesting to propose some.
Regards
Author Response
Por favor, verifique o anexo.
